# 28-day all-cause mortality and associated factors in cancer patients with bacteremia at a peruvian referral center

Yosué Vera[1,2]*, Nicolás Ismael Zamudio[1], Bruno Eduardo Rojas[1], Cesar Cárcamo[3], Pedro Legua[2,4], Brian Delfin[2], Dany Rivera[5], Kathiuska Tutaya[5], Karol Villavicencio[2], Angie Palomino[2], Sissy Monsalve[2], Paola Montenegro[2], Ivan Aguilar[2], Maribel Robles[2], Yenka La Rosa[2], Giuliana Cardenas[2], Frank Young[2], Carlos Rafael Seas[1,2,4‡]

**1** Facultad de Medicina Alberto Hurtado, Universidad Peruana Cayetano Heredia, Lima, Peru, **2** Clínica Oncosalud, AUNA, Lima, Peru, **3** Facultad de Salud Pública, Universidad Peruana Cayetano Heredia, Lima, Peru, **4** Instituto de Medicina Tropical Alexander von Humboldt, Universidad Peruana Cayetano Heredia, Lima, Peru, **5** Laboratorio de Microbiología, AUNA, Lima, Peru

☯ These authors contributed equally to this work.
‡ CRS is a senior author.
* yosue.vera@upch.pe

## Abstract

### Introduction

The global prevalence of cancer has increased in recent decades. Cancer patients are especially vulnerable to invasive infections like bacteremia. In Peru, the rising incidence of bloodstream infections caused by resistant pathogens has become a major public health issue. There is limited information about this complication in the Peruvian cancer population. No previous local studies have assessed mortality or explored the impact of antimicrobial resistance on patient outcomes in this group.

### Methods

This study aimed to evaluate 28-day all-cause mortality and its associated factors among cancer patients with bacteremia at a referral cancer center in Lima. We retrospectively analyzed data from first episodes of bacteremia in hospitalized adult patients between July 2020 and June 2024.

### Results

A total of 293 patients were included in the study. The mean age was 65.4 ± 15.3 years, and 53.9% were female. Most patients had solid tumors (84.3%) and active disease (91.8%), with digestive cancers being the most common (34.1%). The 28-day all-cause mortality rate was 32.1%. Empirical antimicrobial therapy was appropriate in 80.7% of cases. Gram-negative bacteria (GNB) predominated (82.8%), with *Escherichia coli* being the most frequently isolated pathogen (45.7%). Among

**Data availability statement:** All relevant data are within the paper and its Supporting Information files.

**Funding:** The author(s) received no specific funding for this work.

**Competing interests:** The authors have declared that no competing interests exist.

Enterobacteriaceae, 43.6% were extended-spectrum beta-lactamase (ESBL) producers, and 3.3% of GNB were carbapenem-resistant. Multivariable Poisson regression identified the Charlson Comorbidity Index (RR 1.1; 95% CI 1.1–1.2), sepsis (RR 2.3; 95% CI 1.3–3.8), septic shock (RR 3.0; 95% CI 1.9–4.6), respiratory failure (RR 1.6; 95% CI 1.2–2.2), Coagulase-negative *Staphylococci* (CoNS) bacteremia (RR 1.8; 95% CI 1.2–2.6) and primary source (RR 2.8; 95% CI 1.5–5.1) as independent factors associated with increased 28-day mortality.

## Conclusion

At this cancer referral center, one-third of patients with bacteremia died within 28 days. Mortality was primarily caused by the severity of infection, comorbidities and specific bacteremia characteristics, rather than antimicrobial resistance.

## Introduction

Bloodstream infections (BSI) are among the most serious and frequent complications in cancer patients; an estimated 10%–38% of oncology patients develop BSI during their illness [1,2]. These infections occur in both solid and hematologic malignancies, with a higher risk in the latter [3,4], and are most often caused by Gram-negative bacilli [2,5].

Despite advances in care, BSI-related mortality in cancer patients remains high, with 30-day mortality rates of 17%–32% reported in prior studies [3,4,6]. Mortality is primarily associated with cancer status and severity of clinical presentation [2,7,8]. Advanced age, corticosteroid use, polymicrobial infection, and inadequate empirical antibiotic therapy are additional predictors [8–11]. Early mortality (within 48–72 hours of diagnosis) is strongly linked to delayed initiation of active antimicrobial therapy [2,12]. The contribution of antimicrobial resistance is less clear; while some observational studies report higher mortality with multidrug-resistant organisms (MDROs) [12], this has not been consistent across all cancer cohorts with bacteremia [4,9].

In Latin America, few observational studies have examined bacteremia in cancer patients. Despite heterogeneity in their populations, their findings largely mirror global data. Studies including patients with any cancer type in Mexico and Colombia reported 30-day mortality rates of 22% and 25.6%, respectively [2,4]. In Costa Rica, the 30-day mortality rate was 30%, although only patients with solid tumors were included [13]. Among those with hematologic malignancies, a study from Argentina reported a 30-day mortality rate of 17.5% [3]. A retrospective study in Chile, Ecuador, and Peru found a 30-day mortality rate of 15.3% in patients with febrile neutropenia and acute leukemia or lymphoma who developed bacteremia [14]. Although these data may approximate our local reality due to geographical proximity, their restricted populations limit extrapolation to a broader oncologic populations.

Thus, comprehensive data reflecting local epidemiology and patient characteristics are needed to fill knowledge gaps and guide prevention and treatment strategies

in our setting. This study, therefore, aimed to assess 28-day all-cause mortality and its associated factors among cancer patients with bacteremia at a national referral cancer center in Lima.

## Materials and methods

### Study design

A retrospective study was conducted at a single cancer referral center in Lima, Peru. From July 2020 through June 2024, all positive blood culture results were collected. Hospitalized adult patients with any malignancy who experienced their first episode of bacteremia were included. Subsequent episodes of bacteremia in the same individual were not included. Individuals who were lost to follow-up or transferred to another institution within 28 days were excluded. Additionally, patients with synchronous neoplasia (two distinct primary malignancies diagnosed within 6 months) or without histopathological confirmation of cancer were not included in the study.

### Definitions

An episode of bacteremia was defined as 28 days beginning on the date of sampling of at least one positive peripheral blood culture bottle for pathogenic bacteria. If an organism listed as a common commensal (National Healthcare Safety Network) was isolated, it had to be isolated from at least two separate blood cultures taken from different sites, along with clinical signs of infection (fever, chills, or hypotension), to be considered true bacteremia [15,16]. Sepsis and septic shock were diagnosed based on the criteria outlined by the third international consensus for sepsis and septic shock (Sepsis-3) [17]. The Sequential Organ Failure Assessment Score (SOFA) was calculated using clinical and laboratory data obtained closest to the time of culture collection. Empirical therapy was deemed adequate if at least one antibiotic with *in vitro* activity was started before susceptibility results were available. Bacteremia was classified as primary if no secondary sources were identified. It was considered persistent if the same organism was isolated again in a follow-up peripheral blood culture during the same hospital stay. An episode of bacteremia was classified as polymicrobial when two or more bacteria were isolated within the same 72-hour period [8]. The site of acquisition was categorized as nosocomial, community-acquired, or healthcare-associated. Nosocomial bacteremia included patients hospitalized for ≥48 hours in a general ward or intensive care unit before culture collection. Healthcare-associated bacteremia involved patients with any of the following within the past 90 days: hospitalization, intravenous chemotherapy, antibiotic treatment, hemodialysis, wound care, residence in a nursing home or assisted living facility, or the presence of an invasive device within the last 30 days before culture sampling [18]. Advanced cancer was defined as any malignant neoplasm with locally advanced or metastatic disease at the time of study. Active cancer was identified if the patient met any of these criteria: systemic treatment within the previous 6 months, recent tumor resection, end-stage palliative care, recent diagnosis, or evidence of disease progression or recurrence. Immunosuppressive therapy was defined as patients who received any of the following within 30 days before blood culture: systemic chemotherapy, corticosteroids, monoclonal antibodies, tyrosine kinase inhibitors, or immunotherapy. Past antibiotic use was defined as having received any dose of antibiotics in the previous 30 days. Prior hospitalization meant patients who had been hospitalized for at least 48 hours within the last three months before the bacteremia episode. Comorbidities were assessed with the Charlson Comorbidity Index (CCI), a validated score that estimates 10-year survival probability [19].

### Microbiology

Multidrug-resistant Gram-negative bacteria (GNB) were defined as non-susceptible to at least one antibiotic in three or more different classes [20]. *Stenotrophomonas maltophilia* was included in this group due to its inherent resistance traits. Strains resistant to all tested antibiotic classes, except one or two, were categorized as extremely resistant GNB [20]. Difficult-to-treat resistant (DTR) GNB were those that show resistance or intermediate susceptibility to every

antibiotic in three specific classes: beta-lactams, carbapenems, and fluoroquinolones [21]. The following Enterobacteriaceae species were considered AmpC cephalosporinase hyperproducers: *Enterobacter cloacae*, *Klebsiella aerogenes* and *Citrobacter freundii* [22]. Species at risk of AmpC hyperproduction that exhibited resistance to third-generation cephalosporins but susceptibility to cefepime were considered constitutive AmpC cephalosporinase hyperproducers [22]. Extended-spectrum beta-lactamase (ESBL) producers were identified by non-susceptibility to either ceftazidime, cefotaxime, ceftriaxone, or cefepime.

## Procedures

Laboratory technicians collected blood samples from the institution. Four peripheral samples, each with 5−10 mL of blood, were drawn for two blood culture sets each taken from different sites. Each set contained two bottles with culture media—one for anaerobes and one for aerobes. The growth media used were BACT/ALERT® SA Standard Aerobic (bioMérieux, France) and BACT/ALERT® SN Standard Anaerobic (bioMérieux, France), both containing 40 mL of Supplemented Tryptic Soy Broth. Samples were processed using the BACT/ALERT VIRTUO microbial detection system (bioMérieux, France), which employs a colorimetric method for growth detection. These were incubated in a media containing antibiotic remover for a maximum of 96 hours. The positive samples were subsequently incubated for another 18–24 hours in conventional growth media for further testing. Bacterial identification and susceptibility testing were performed using the automated VITEK 2 COMPACT system (bioMérieux, France), which utilizes cards tailored to the specific pathogen type. Susceptibility results were interpreted according to CLSI guidelines. Carbapenemases were detected via chromatographic tests: RAPIDEC® CARBA NP (bioMérieux, France) or OKNVI RESIST-5 (Coris BioConcept, Belgium).

## Statistical analysis

Variables with more than 10% missing data were excluded from the main analysis. Bivariate analysis was conducted for both the primary and secondary outcomes. Chi-square test or Fisher's exact test was used for categorical variables. After assessing normality with the Shapiro-Wilk test, non-categorical variables were analyzed using the t-test or the Mann-Whitney U test. Two multivariable analyses were performed for both the primary and secondary outcomes. Variables included in the multivariable model were those with a p-value less than 0.20 in the bivariate analysis and/or those clinically relevant. The Poisson regression model was employed for both multivariable analyses. All variables were considered statistically significant at a p-value of less than 0.05. The list of variables included in the bivariate and multivariable analyses is shown in S1 and S2 Tables in S1 File.

## Ethical aspects

The project was approved by the Institutional Research Ethics Committee of Universidad Peruana Cayetano Heredia and Clínica Oncosalud. No access to patients' personal data was granted, as all information was coded for the study, ensuring that the authors had no access to participants' identifying details. Data were collected retrospectively between August 1st and December 15th, 2024. Access to the database was restricted to the principal investigators.

## Results

### Study population

A total of 490 positive blood cultures were collected between July 2020 and June 2024. Of these, 61 were deemed contaminants, 39 were from non-hospitalized patients, 76 involved recurrent episodes of bacteremia, and 21 were excluded for other reasons. The study included 293 patients experiencing their first episode of bacteremia (Fig 1).

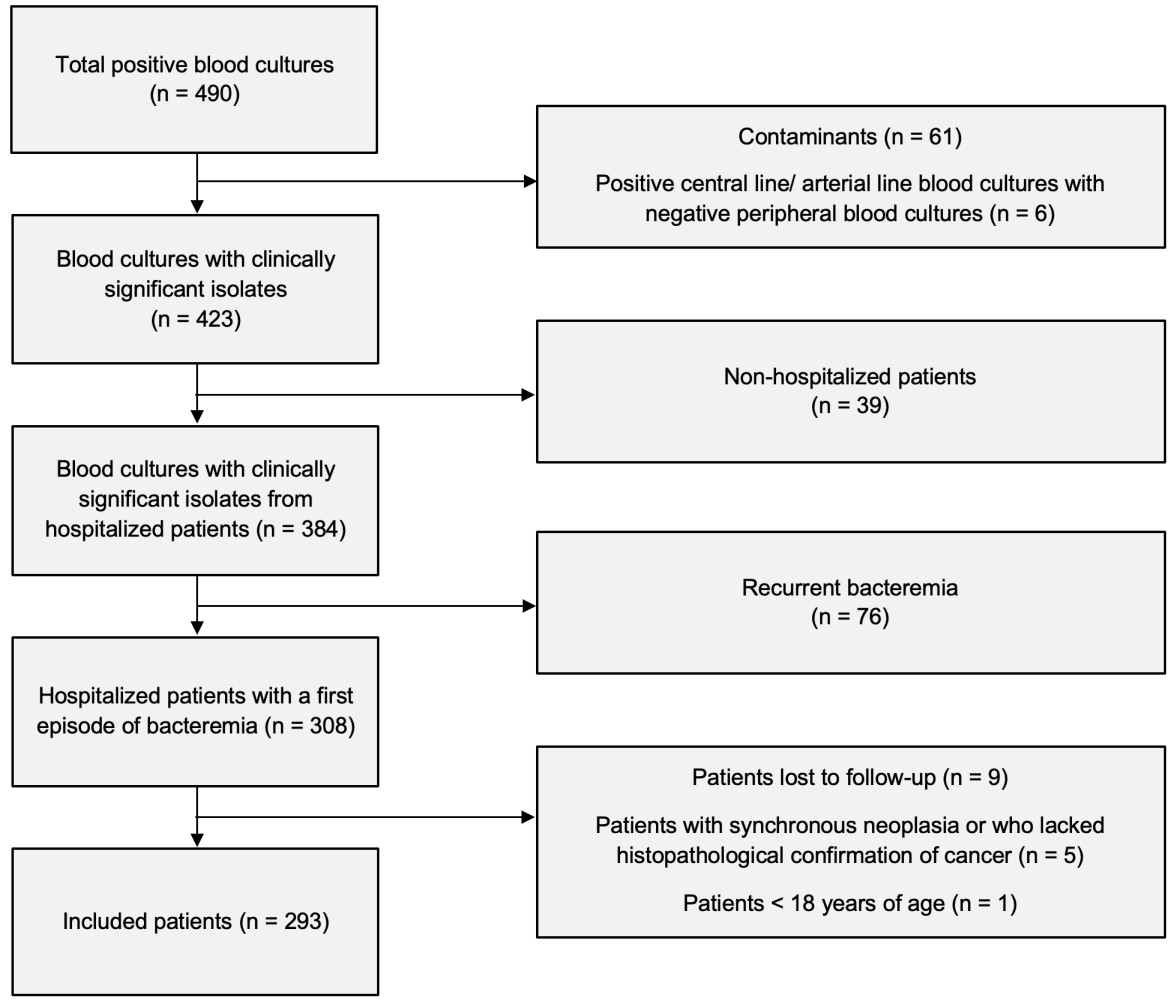

**Fig 1. Flowchart of patient selection.**

## Clinical and demographic characteristics

Solid tumors represented the most common underlying malignancy, accounting for 84.3% of cases. Most patients (91.8%) had active disease, and 80.7% had been diagnosed with advanced-stage cancer at the time of infection. At admission, 49.8% of patients presented with sepsis, while 33.8% experienced septic shock. Additionally, antibiotic use in the previous month was documented for 46.8% of cases, and 67.9% of patients had experienced a recent hospitalization. Other clinical and demographic characteristics are summarized in Table 1.

Meropenem was the most frequently used empiric antibiotic (59.0%) and was significantly associated with higher rates of appropriate empiric therapy. For further information, refer to S3 and S4 Tables in S1 File.

## Bacteremia features and microbiological findings

The characteristics of bacteremia are listed in Table 2. Secondary bacteremia accounted for 53.9%, with urinary tract infections being the most common source at 23.9%. Nine cases (3.1%) were polymicrobial: five exclusively involved GNB, and four involved both GPB and GNB. Persistent bacteremia was assessed in 181 patients (61.8%) and was present in 17 (9.4%).

**Table 1. Clinical and demographic characteristics of 293 patients with bacteremia\*†.**

| Characteristic | Total |
| --- | --- |
| Age, years, mean±SD‡ | 65.4±15.3 |
| Female sex | 158 (53.9) |
| Charlson Comorbidity Index, mean±SD‡ | 7.0±2.8 |
| Hematologic malignancy | 46 (15.7) |
| Solid tumors | 247 (84.3) |
| Active cancer | 269 (91.8) |
| Advanced neoplasm, n=275 | 222 (80.7) |
| Empirical antibiotic therapy | 285 (97.3) |
| Adequate empirical antibiotic therapy, n=285 | 230 (80.7) |
| Sepsis | 146 (49.8) |
| Septic shock | 99 (33.8) |
| Respiratory failure on admission§ | 35 (12.0) |
| Severe neutropenia, n=287 | 29 (10.1) |
| Prior major surgery | 95 (32.4) |
| Immunosuppressive therapy | 129 (44.0) |
| Prior antibiotic use | 137 (46.8) |
| Prior hospitalization | 199 (68.0) |
| Central venous catheter | 50 (17.1) |
| Port-a-Cath | 102 (34.8) |

\*Calculated on 293 patients unless otherwise specified.

†Values are number (%) unless otherwise noted.

‡SD: standard deviation.

§Within 48 hours of blood culture sampling.

**Table 2. Bacteremia characteristics\*†.**

| Characteristic | Total |
| --- | --- |
| Bacteremia source | |
| Primary | 135 (46.1) |
| Secondary | 158 (53.9) |
| Urinary | 70 (23.9) |
| Biliary | 34 (11.6) |
| Abdominal | 20 (6.8) |
| Respiratory | 19 (6.5) |
| Skin and soft tissue | 9 (3.1) |
| Other | 6 (2.1) |
| Place of bacteremia acquisition | |
| Community | 41 (14.0) |
| Nosocomial | 48 (16.4) |
| Healthcare-associated | 204 (69.6) |
| Polymicrobial bacteremia | 9 (3.1) |
| Persistent bacteremia, n=181 | 17 (9.4) |

\*Calculated on 293 patients unless otherwise specified.

†Values are numbers (%).

A total of 302 isolates were retrieved, with GNB predominating (82.8%). The most frequently isolated species was *Escherichia coli* (45.7%), followed by *Klebsiella pneumoniae* (14.2%). The different bacterial species isolated are shown in Table 3. Among GPB, Coagulase-negative *Staphylococci* (CoNS) were the most common (7.0%), with *Staphylococcus epidermidis* accounting for most cases (S5 Table in S1 File).

*Acinetobacter baumannii* bacteremia had the highest mortality rate among all pathogens (75.0%), followed by CoNS (61.9%) and *P. aeruginosa* (40%). CoNS were the only species with a statistically significant higher mortality rate at 28 days compared to other pathogens (61.9% vs. 29.8%, $p < 0.05$). 28-day mortality rates by bacterial species are shown in S6 Table in S1 File.

## Antimicrobial resistance phenotypes

Resistance phenotypes are presented in Table 4. BSI caused by GNB showed a higher rate of antimicrobial resistance (AMR) compared to those caused by GPB (66.1% vs. 45.8%, $p < 0.05$). ESBL-mediated resistance to third-generation cephalosporins was the most common resistant phenotype in this cohort. About one-third of all bacteremia episodes were caused by ESBL-producing Enterobacteriaceae. Among all Enterobacteriaceae isolates, 43.6% were ESBL producers.

**Table 3. Distribution of bacterial isolates*†.**

| Classification | Total |
|---|---|
| Gram-positive | 52 (17.2) |
| Coagulase-negative *Staphylococci* | 21 (7.0) |
| *Staphylococcus aureus* | 12 (4.0) |
| *Enterococcus faecalis* | 11 (3.6) |
| *Streptococcus pyogenes* | 2 (0.7) |
| *Streptococcus anginosus* | 2 (0.7) |
| *Streptococcus agalactiae* | 1 (0.3) |
| *Streptococcus constellatus* | 1 (0.3) |
| *Enterococcus faecium* | 1 (0.3) |
| *Listeria monocytogenes* | 1 (0.3) |
| Gram-negative | 250 (82.8) |
| Fermenters | 215 (71.2) |
| *Escherichia coli* | 138 (45.7) |
| *Klebsiella pneumoniae* | 43 (14.2) |
| *Serratia marcescens* | 9 (3.0) |
| *Enterobacter cloacae* | 8 (2.7) |
| *Proteus mirabilis* | 4 (1.3) |
| *Salmonella enterica* | 4 (1.3) |
| Other | 9 (3.0) |
| Non-fermenters | 29 (9.6) |
| *Pseudomonas aeruginosa* | 20 (6.6) |
| *Acinetobacter baumannii* | 4 (1.3) |
| *Stenotrophomonas maltophilia* | 3 (1.0) |
| Other | 2 (0.7) |
| Anaerobes | 6 (2.0) |
| *Bacteroides sp.* | 6 (2.0) |

*Calculated on 302 bacterial isolates unless otherwise specified.

†Values are number (%).

**Table 4. Antimicrobial resistance profiles\*†.**

| Phenotype | Total |
|---|---|
| Gram-negative bacteria | 245 (83.6) |
| Carbapenem-resistant, n = 245 | 8 (3.3) |
| Multidrug-resistant, n = 245 | 100 (40.8) |
| Extensively drug-resistant, n = 245 | 34 (13.9) |
| Difficult-to-treat resistance, n = 245 | 5 (2.0) |
| Extended-spectrum β-lactamase-producing Entero-bacteriaceae, n = 211 | 92 (43.6) |
| AmpC cephalosporinase-hyperproducing Entero-bacteriaceae, n = 211 | 24 (11.4) |
| Constitutive AmpC cephalosporinase hyperproducers, n = 211 | 3 (1.4) |
| Gram-positive bacteria | 52 (17.8) |
| *Staphylococcus aureus* | 12 (4.1) |
| Methicillin-resistant, n = 12 | 0 (0) |
| Coagulase-negative *Staphylococci* | 21 (7.2) |
| Methicillin-resistant, n = 21 | 15 (71.4) |
| *Enterococcus sp.* | 12 (4.1) |
| Ampicillin-resistant | 1 (8.3) |

*Calculated on 293 patients unless otherwise specified.

†Values are number (%).

For *E. coli* and *K. pneumoniae,* the proportions were higher, with 51.6% and 51.2% of isolates producing ESBLs, respectively. Fatality rates did not differ between ESBL-positive and ESBL-negative Enterobacteriaceae bacteremia (see S7 Table in S1 File). Only eight Gram-negative isolates were carbapenem-resistant, with a nonsignificant higher mortality rate (37.5%) compared to susceptible strains. All *Acinetobacter baumannii* isolates were carbapenem-resistant; Class D carbapenemases (OXA-23, OXA-24, OXA-25, and OXA-58) were detected in two, both from patients with intensive care unit (ICU)-acquired bacteremia. Among 20 *Pseudomonas aeruginosa* isolates, two were carbapenem-resistant: one produced a Class B carbapenemase (metallo-β-lactamase), and the other one showed probable resistance due to loss of the OprD2 porin or overproduction of efflux pumps, based on susceptibility patterns. The only two carbapenem-resistant *K. pneumoniae* strains produced OXA-48 carbapenemase. Among Gram-positive isolates, the most common resistant phenotype was Methicillin-resistant CoNS (MR-CoNS) (15/21). All *Staphylococcus aureus* strains were methicillin-susceptible, and no vancomycin-resistant GPB were identified.

## Clinical outcomes

The 28-day all-cause mortality rate was 32.1%. The 48-hour and 90-day all-cause mortality rates were 9.2% and 48.6%, respectively; 20.8% of patients were admitted to the ICU within 48 hours of blood culture sampling. The clinical outcomes are shown in Table 5.

## Factors associated with mortality

Table 6 summarizes the bivariate and multivariable analyses for 28-day all-cause mortality. Independent predictors of mortality included a higher CCI, the presence of sepsis and septic shock on admission, respiratory failure, CoNS bacteremia and the bacteremia source. Septic shock had the strongest association with mortality (RR, 3.0; 95% CI, 1.9–4.6; p < 0.001). Among CCI components, heart failure and solid tumors (localized or metastatic) were significantly associated

**Table 5. Clinical outcomes\*†.**

| Outcome | Total | 95% CI‡ |
|---|---|---|
| All-cause mortality | | |
| 48-hour | 27 (9.2) | [6.4–13.1] |
| 28-day | 94 (32.1) | [27.0–37.6] |
| 90-day§ | 142 (48.6) | [42.8–54.2] |
| ICU admission within 48 hours | 61 (20.8) | [16.6–25.8] |
| Readmission‖ | 112 (56.3) | [32.9–43.9] |
| Bacteremia-associated, n = 112 | 20 (17.9) | [11.8–26.3] |
| Length of stay, days, median with IR¶ | | |
| Total length of stay | 12 [8–20] | |
| Length of stay due to bacteremia | 10 [6–17] | |

\*Calculated on 293 patients unless otherwise specified.

†Values are number (%) unless otherwise noted.

‡CI = confidence interval.

§n = 292, 1 lost to follow-up.

‖n = 199, deceased patients at 28 days were not included.

¶IR = interquartile ranges.

**Table 6. Bivariate and multivariable analysis of factors associated with 28-day all-cause mortality\*†.**

| | 28-day all-cause mortality | | Bivariate | Multivariable | |
|---|---|---|---|---|---|
| Characteristics | Alive (n = 199) | Deceased (n = 94) | p-value | RR§ [95% CI‖] | p-value |
| Charlson Comorbidity Index, mean ± SD¶ | 6.6 ± 2.8 | 7.9 ± 2.7 | < 0.001\*\* | 1.1 [1.1-1.2] | < 0.001 |
| Bacteremia source | | | < 0.001†† | | |
| Urinary | 61 (30.7) | 9 (9.6) | | 1.0 | |
| Primary | 78 (39.2) | 57 (60.6) | | 2.8 [1.5-5.1] | 0.001 |
| Other | 60 (30.2) | 28 (29.8) | | 2.0 [1.0-3.7] | 0.042 |
| Respiratory failure | 33 (16.6) | 50 (53.2) | < 0.001†† | 1.6 [1.2-2.2] | 0.005 |
| Coagulase-negative *Staphylococci* | 8 (4.0) | 13 (13.8) | 0.002†† | 1.8 [1.2-2.6] | 0.003 |
| Sepsis | 74 (37.2) | 72 (76.6) | < 0.001†† | 2.3 [1.3-3.8] | 0.002 |
| Septic shock | 46 (23.1) | 53 (56.4) | < 0.001†† | 3.0 [1.9-4.6] | < 0.001 |

\*Calculated on 293 patients unless otherwise specified.

†Values are number (%) unless otherwise noted.

‡Adjusted for age and sex.

§RR: relative risk.

‖CI: robust confidence interval.

¶SD: standard deviation.

\*\*Student's *t*-test.

††Chi-square test.

with increased risk (S8 Table in S1 File). Each additional CCI point was associated with an 10% increase in mortality. The primary source was found to be an important predictor of mortality when compared with the urinary source. CoNS as a causative agent of bacteremia were found to be predictors of mortality. No specific resistant phenotype was independently associated with mortality.

Early (48-hour) mortality predictors are shown in Table 7. Appropriate empiric antibiotic therapy was associated with a significantly lower risk of early death (RR 0.3; 95% CI, 0.2–0.6; p = 0.001). Conversely, primary source, sepsis, and septic shock were significantly associated with early mortality.

## Discussion

In this cancer referral center, the 28-day all-cause mortality rate among patients who developed bacteremia was 32.1%. The primary independent factors associated with mortality were infection severity, comorbidities, and certain specific bacteremia phenotypes. ESBL-producing Enterobacteriaceae were the most common resistant pathogens, responsible for nearly half of the bacteremia cases. Contrary to expectations, antibiotic resistance was not independently linked to increased mortality.

The observed mortality rate in our cohort is among the highest reported for cancer patients with bacteremia. Previous studies have reported mortality rates ranging from 14.9% to 32%, depending on the predominant malignancy type in each cohort [2,4,7,9,23]. Compared with cohorts from the same geographic area, our center's mortality rate remains the highest [2–4,24]. This may be partly due to the poor baseline prognosis of our cohort: we included a larger proportion of elderly patients, and the CCI score was significantly higher, even though the proportion of individuals with active or advanced cancer was similar to that of other cohorts [2,4]. These findings highlight the unfavorable baseline expected survival of the patients included in our study.

Pre-infectious conditions determined mortality in this cohort. Comorbidities, including cancer, appear to partially determine outcomes during bacteremia. A higher CCI was independently linked to increased mortality, consistent with Ha et al., who reported this association in cancer patients with *E. coli* bacteremia [23]. Among the CCI components, we found statistically significant differences in 28-day survival between patients with local and metastatic neoplasms. Cancer burden has consistently been identified as a marker of higher mortality during bacteremia episodes [2–4,7,9]. Likewise, infection severity was a fundamental predictor of mortality. Septic shock at presentation was the strongest driver of mortality in our cohort. Several studies have consistently identified sepsis and septic shock as predictive of higher mortality [3,6–8,23,25].

**Table 7. Bivariate and multivariable analysis of factors associated with 48-hour all-cause mortality[*][†].**

| Characteristics | 48-hour all-cause mortality | | Bivariate | Multivariable[‡] | |
|---|---|---|---|---|---|
| | Alive (n = 266) | Deceased (n = 27) | p-value | RR[§] [95% CI[‖]] | p-value |
| Adequate empiric antibiotic therapy | 214 (80.5) | 16 (59.3) | 0.011[¶] | 0.3 [0.2-0.6] | 0.001 |
| Primary source of bacteremia | 113 (42.5) | 22 (81.5) | < 0.001[¶] | 3.4 [1.3-8.9] | 0.013 |
| Respiratory failure | 23 (8.7) | 12 (44.4) | < 0.001[¶] | 2.5 [1.3-4.6] | 0.006 |
| Sepsis | 43 (16.2) | 4 (14.8) | < 0.001[¶] | 15.8 [1.9-130.3] | 0.010 |
| Septic shock | 77 (29.0) | 22 (81.5) | < 0.001[¶] | 29.9 [4.3-208.2] | 0.001 |

*Calculated on 293 patients unless otherwise specified.

†Values are number (%) unless otherwise noted.

‡Adjusted for age and sex.

§RR: relative risk.

‖CI: robust confidence interval.

¶Chi-square test.

Our findings repeatedly emphasize the importance of individual baseline status on clinical outcomes, as sepsis is known to result from an aberrant host immune response to infection [26]. Additionally, respiratory failure was independently linked to higher mortality. No studies have specifically reported this association. Ha et al. noted that the need for mechanical ventilation predicted increased mortality [23]. In our cohort, respiratory failure could have resulted from either cancer progression or multiorgan failure caused by severe infection. This highlights the importance of comorbidities and infection severity in cancer patients with bacteremia. Interestingly, bacteremia caused by CoNS was identified as an independent risk factor for higher mortality compared with other pathogens. This could be attributed to the high rate of methicillin resistance in our center's CoNS isolates (15/21), which led to lower rates of appropriate empiric therapy in this subgroup (~50%). No previous studies have identified CoNS bacteremia as a predictor of mortality in cancer patients. In a nationwide Korean study, Kang et al. found *S. aureus* to be independently associated with increased mortality in cancer patients with bacteremia [6]. This association was not observed in our center, as mortality among patients with *S. aureus* bacteremia did not differ significantly from that associated with other pathogens, possibly due to the absence of Methicillin-resistant *Staphylococcus aureus* (MRSA) strains. The primary infection source was also identified as independently associated with elevated mortality compared with the urinary source. This finding aligns with the results of Cuervo et al., who observed higher mortality in a Colombian cohort of cancer patients with *S. aureus* bacteremia when a primary focus of infection was present [27]. Similarly, another Colombian study, not limited to cancer patients, identified non-urinary sources of bacteremia as being associated with increased mortality [28]. These observations suggest that specific bacteremia phenotypes—such as the causative agent or the source—may influence prognosis. A sensitivity analysis excluding patients with hematological malignancies and polymicrobial bacteremia was performed to evaluate the consistency of our findings (S9 Table in S1 File). In this subgroup, factors associated with 28-day all-cause mortality did not substantially differ, thereby strengthening the internal validity of our findings.

Unlike previous reports, no association between antimicrobial resistance and mortality was found at this center. This may be partly attributable to the adequate rate of appropriate empirical therapy, particularly the frequent use of carbapenems, since ESBL-producing Enterobacteriaceae were the predominant antimicrobial-resistant phenotype in our cohort. Ha et al. demonstrated that ESBL presence was associated with increased mortality in cancer patients with bacteremia [23]. Similarly, Gudiol et al. reported higher death rates with Gram-negative MDR organisms [29]. While carbapenem-resistant GNB exhibited increased mortality compared to susceptible strains, this was not identified as a risk factor for mortality in the multivariable analysis. Increased all-cause mortality from this strains has also been previously reported in hematologic cancer patients with bacteremia [3]. A type II error might account for the lack of association, possibly due to the low incidence of carbapenem-resistant strains at this center.

Antimicrobial resistance rates at our center generally align with the latest national data. ESBL-producing Enterobacteriaceae were the dominant AMR phenotype in our cohort and are recognized as a national issue. A multicenter study by Krapp et al. across various regions of Peru found a concerning 59.2% of third-generation cephalosporin-resistant GNB isolated from blood samples in public hospitals [30]. The lower ESBL-positive rate observed in our cohort may be due to a more robust infection-control program, as it was implemented at a private center with economic advantages. We identified eight carbapenem-resistant GNB, representing 3.27% of the total an even lower proportion compared to other regional studies [3,4,30]. No carbapenem-resistant *E. coli* were detected, which remains a rare strain in Peru [30]. Notably, no cases of MRSA bacteremia were identified, which opposes local data reporting a 36.1% prevalence in BSI [31]. To date, no studies have examined differences in AMR between public and private institutions in Peru. The lower AMR rates observed in our center likely reflect the socioeconomic gap and differences in resource availability between these types of health service providers. Data from Brazil suggest that private institutions have better adherence to sepsis protocols, including earlier recognition, treatment, and timely blood culture sampling, which has been linked to better clinical outcomes [32]. Our study suggests that AMR may vary across different socioeconomic contexts within our country; however, more studies are needed to confirm these observations.

The main limitation of this study is its retrospective design. The data analyzed were collected at a single private center and were limited to individuals with medium-to-high socioeconomic status, reducing external validity. Additionally, the short study period limits statistical power for certain variables. Using all-cause mortality as the primary outcome may have underestimated the direct impact of bacteremia on patient outcomes. Nevertheless, this is the first study to evaluate the impact of bacteremia among individuals with cancer in Peru. Globally, our findings contribute to the existing evidence on this common infectious complication in people with cancer.

## Conclusions

In this group of individuals, mostly with solid tumors and bacteremia, approximately one-third died within 28 days of diagnosis. The severity of the infection, comorbidities, and specific infection-related characteristics primarily influenced mortality. Antimicrobial-resistant bacteria were not associated with worse outcomes in this center.

## Supporting information

**S1 File. Supplemental tables 1–9.** Additional methodological data and analyses, including sensitivity analyses for 48-hour and 28-day all-cause mortality.
(PDF)

**S1 Data. Supplementary dataset 1.** De-identified patient-level data used for the analysis of 28-day all-cause mortality and associated factors in cancer patients with bacteremia.
(XLSX)

## Author contributions

**Conceptualization:** Yosué Vera, Nicolás Ismael Zamudio, Carlos Rafael Seas.

**Data curation:** Yosué Vera, Nicolás Ismael Zamudio, Bruno Eduardo Rojas, Dany Rivera, Kathiuska Tutaya, Carlos Rafael Seas.

**Formal analysis:** Yosué Vera, Nicolás Ismael Zamudio, Cesar Cárcamo.

**Investigation:** Yosué Vera, Nicolás Ismael Zamudio, Pedro Legua, Brian Delfin, Dany Rivera, Kathiuska Tutaya, Karol Villavicencio, Angie Palomino, Sissy Monsalve, Paola Montenegro, Ivan Aguilar, Maribel Robles, Yenka La Rosa, Giuliana Cardenas, Frank Young, Carlos Rafael Seas.

**Methodology:** Yosué Vera, Nicolás Ismael Zamudio, Bruno Eduardo Rojas, Cesar Cárcamo, Carlos Rafael Seas.

**Resources:** Paola Montenegro, Ivan Aguilar, Maribel Robles, Yenka La Rosa, Giuliana Cardenas, Frank Young, Carlos Rafael Seas.

**Supervision:** Yosué Vera, Cesar Cárcamo, Pedro Legua, Carlos Rafael Seas.

**Validation:** Yosué Vera, Nicolás Ismael Zamudio, Bruno Eduardo Rojas, Cesar Cárcamo, Brian Delfin, Karol Villavicencio, Angie Palomino, Sissy Monsalve, Ivan Aguilar, Maribel Robles, Yenka La Rosa, Giuliana Cardenas, Frank Young, Carlos Rafael Seas.

**Writing – original draft:** Yosué Vera, Nicolás Ismael Zamudio, Bruno Eduardo Rojas, Pedro Legua, Brian Delfin, Dany Rivera, Carlos Rafael Seas.

**Writing – review & editing:** Yosué Vera, Nicolás Ismael Zamudio, Bruno Eduardo Rojas, Cesar Cárcamo, Pedro Legua, Brian Delfin, Dany Rivera, Kathiuska Tutaya, Karol Villavicencio, Angie Palomino, Sissy Monsalve, Paola Montenegro, Ivan Aguilar, Maribel Robles, Yenka La Rosa, Giuliana Cardenas, Frank Young, Carlos Rafael Seas.

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
