## [Decision Letter · Decision Letter 0]

19 Jan 2026

PONE-D-25-6472028-Day All-Cause Mortality and Related Factors in Cancer Patients with Bacteremia at a Peruvian Referral CenterPLOS One

Dear Dr. Vera,

Thank you for submitting your manuscript to PLOS ONE. After careful consideration, we feel that it has merit but does not fully meet PLOS ONE’s publication criteria as it currently stands. Therefore, we invite you to submit a revised version of the manuscript that addresses the points raised during the review process.

We look forward to receiving your revised manuscript.

Kind regards,

Mabel Kamweli Aworh, DVM, MPH, PhD. FCVSN

Academic Editor

PLOS One

Journal Requirements:

Reviewers' comments:

Reviewer's Responses to Questions

**Comments to the Author**

1. Is the manuscript technically sound, and do the data support the conclusions?

Reviewer #1: Yes

Reviewer #2: Yes

Reviewer #3: Yes

2. Has the statistical analysis been performed appropriately and rigorously? 

Reviewer #1: Yes

Reviewer #2: Yes

Reviewer #3: Yes

3. Have the authors made all data underlying the findings in their manuscript fully available?

Reviewer #1: No

Reviewer #2: Yes

Reviewer #3: Yes

4. Is the manuscript presented in an intelligible fashion and written in standard English?

Reviewer #1: Yes

Reviewer #2: Yes

Reviewer #3: Yes

5. Review Comments to the Author

Reviewer #1: Kudos to the researchers for this important work.

I think your statistical analysis section will benefit from added clarity. Consider the following:

1. Be explicit as to whether you performed multivariate or multivariable analysis. If your outcomes, 28-day and 48-hour mortality were analyzed separately, then that is two multivariable analysis.

2. Specify which variables were included in each analysis and test.

3. In line 155, you mentioned that variables included in the model were those with a p-value less than 0.20. Please clarify which test or analysis that this p-value comes from.

4. While Poisson regression is appropriate for estimating RR in this cohort study, please clarify whether robust standard errors were applied to ensure valid confidence intervals and p-values given binary outcome.

Reviewer #2: The authors did a good job. Just some clarifications needed though. Some aspects of their methodology need clearer definitions and parameters.

I would also suggest subgroup analysis of more potential confounders like grade/stage of the advanced cancer, presence of neutropenia. This may influence their findings and the comparative conclusions made in the discussion.

Reviewer #3: The study is timely and clinically significant, however it requires clearer methodological details, stronger statistical transparency and sensitivity analysis to support the conclusions.

Suggested Revisions:

Introduction

- Streamline the Introduction to reduce general background and emphasize the specific evidence gaps in Latin America

Methods

-Clarify inclusion/exclusion criteria

-Provide a detailed list of the organisms classified as commensals

Statistical analysis:

-Please justify the choice of Poisson regression for binary outcomes and indicate whether robust variance was applied.

- Need to include a detailed description of the variable selection process and how missing data were handled

-Consider adding sensitivity analyses, for example, stratifying by malignancy type or excluding polymicrobial cases to strengthen the robustness of the findings.

Result

-Ensure denominators are clearly stated in all tables

-In all tables, consider using one-decimal points for all proportions

-Table 2; correct truncation and realign table for clarity

e,g “Nosocomial 48 (16.38” , the closing parenthesis and percentage are incomplete,

- Ensure all denominators are consistent

Discussion

- Please expand the discussion to acknowledge the limitations of using all‑cause mortality

- Compare findings to other Latin American cohorts

- Discuss how private referral center context may influence results and generalizability, for example, differences in patient demographics, resource availability, or treatment protocols compared to public institutions

6. PLOS authors have the option to publish the peer review history of their article (what does this mean?). If published, this will include your full peer review and any attached files.

Reviewer #1: No

Reviewer #2: **Yes:**Oluchi I Ndulue

Reviewer #3: **Yes:**Elizabeth B. Adedire

---

## [Author Response · Author response to Decision Letter 1]

23 Mar 2026

Dear Editors and Reviewers,

We sincerely thank you for your careful evaluation of our manuscript and for your insightful, constructive comments. Your suggestions have greatly improved the clarity, rigor, and overall quality of the manuscript. We have carefully addressed each comment below. All changes in the revised manuscript are highlighted using track changes.

As the reviewers suggested, we have redone the statistical analysis. For that purpose, we invited Dr. Cesar Carcamo to assist us. We have added him to the list of co-authors and request your approval.

Comments to the Author

1. Is the manuscript technically sound, and do the data support the conclusions?

Reviewer #1: Yes

Reviewer #2: Yes

Reviewer #3: Yes

2. Has the statistical analysis been performed appropriately and rigorously?

Reviewer #1: Yes

Reviewer #2: Yes

Reviewer #3: Yes

3. Have the authors made all data underlying the findings in their manuscript fully available?

The PLOS Data policy requires authors to make all data underlying the findings described in their manuscript fully available without restriction, with rare exceptions (please refer to the Data Availability Statement in the manuscript PDF file). The data should be provided as part of the manuscript or its supporting information, or deposited to a public repository. For example, in addition to summary statistics, the data points behind means, medians and variance measures should be available. If there are restrictions on publicly sharing data—e.g. participant privacy or use of data from a third party—those must be specified.

Reviewer #1: No

Reviewer #2: Yes

Reviewer #3: Yes

4. Is the manuscript presented in an intelligible fashion and written in standard English?

Reviewer #1: Yes

Reviewer #2: Yes

Reviewer #3: Yes

5. Review Comments to the Author

Reviewer #1: Kudos to the researchers for this important work.

+I think your statistical analysis section will benefit from added clarity. Consider the following:

1. Be explicit as to whether you performed multivariate or multivariable analysis. If your outcomes, 28-day and 48-hour mortality were analyzed separately, then that is two multivariable analysis.

Thank you for your suggestion. We performed two multivariable analyses, which are now specified in lines 205-206. In addition, the term multivariate has been replaced with multivariable throughout the manuscript in accordance with your recommendation.

2. Specify which variables were included in each analysis and test.

Thank you for your suggestion. The list of variables included in each analysis is provided in two supplementary tables: one for the primary outcome and one for the secondary outcome. These are referenced in lines 210–211 of the Statistical Analysis section of the manuscript.

3. In line 155, you mentioned that variables included in the model were those with a p-value less than 0.20. Please clarify which test or analysis that this p-value comes from.

Thank you for your suggestion. Clinically relevant variables with p-values <0.20 in the bivariate analysis were included in the multivariable models, and the statistical tests used are described in lines 202–205. We have also specified the statistical test performed for each variable in Tables 6 and 7.

4. While Poisson regression is appropriate for estimating RR in this cohort study, please clarify whether robust standard errors were applied to ensure valid confidence intervals and p-values given binary outcome.

Thank you for your suggestion. Both multivariable Poisson regression analyses were rerun with robust standard errors. The results are shown in Tables 6 and 7. The discussion and conclusions have been adjusted to account for the new results.

Reviewer #2: The authors did a good job. Just some clarifications needed though. Some aspects of their methodology need clearer definitions and parameters.

I would also suggest subgroup analysis of more potential confounders like grade/stage of the advanced cancer, presence of neutropenia. This may influence their findings and the comparative conclusions made in the discussion.

Thank you for your suggestion. We performed a sensitivity analysis excluding individuals with hematologic malignancies and polymicrobial bacteremia; the results are presented in Supplementary Table 9. The Discussion section has been revised accordingly (See lines 356-360).

Reviewer #3: The study is timely and clinically significant, however it requires clearer methodological details, stronger statistical transparency and sensitivity analysis to support the conclusions.

Suggested Revisions:

Introduction

Streamline the Introduction to reduce general background and emphasize the specific evidence gaps in Latin America -

Thank you for your suggestion. We reduced the background and expanded the discussion of Latin American precedents relevant to the topic. Given the limited data in our region, we further described the articles most similar to our work. The introduction has been revised (see lines 54-79).

Methods

-Clarify inclusion/exclusion criteria

-Provide a detailed list of organisms classified as commensals

Thank you for your suggestion. The inclusion and exclusion criteria section has been expanded to improve clarity and transparency. The list of organisms classified as commensals is available in a freely accessible document, and the link is provided in Reference 18.

Statistical analysis:

- Please justify the choice of Poisson regression for binary outcomes and indicate whether robust variance was applied.

We consider the Poisson regression model appropriate for this study because it estimates relative risks. The high frequency of the primary outcome ensures accurate estimates. Robust confidence intervals have been calculated in this updated version of the analyses.

- Need to include a detailed description of the variable selection process and how missing data were handled.

Thank you for your suggestion. Variables with more than 10% missing data were excluded from the bivariate analysis. Variables for the multivariable analysis were selected based on a p-value <0.20 in the bivariate analysis. We have revised the Statistical Analysis section accordingly (see lines 202 and 207-208).

-Consider adding sensitivity analyses, for example, stratifying by malignancy type or excluding polymicrobial cases to strengthen the robustness of the findings.

Thank you for your suggestion. A sensitivity analysis excluding individuals with hematologic malignancies and polymicrobial bacteremia was performed, and the results are presented in Supplementary Table 9.

Result

-Ensure denominators are clearly stated in all tables.

-In all tables, consider using one-decimal points for all proportions

-Table 2; correct truncation and realign table for clarity

e,g “Nosocomial 48 (16.38” , the closing parenthesis and percentage are incomplete,

- Ensure all denominators are consistent

Thank you very much for your helpful suggestions. We have now included denominators in all tables and reformatted all proportions to one decimal place, as recommended. Table 2 has been realigned for clarity, and the truncation error has been corrected. We carefully reviewed all tables to ensure consistent denominators throughout the manuscript.

Discussion

- Please expand the discussion to acknowledge the limitations of using all‑cause mortality

Thank you for your suggestion. We have acknowledged limitations of using all-cause mortality in the discussion section (see lines 472 - 473).

- Compare findings to other Latin American cohorts

Thank you for your thoughtful suggestion. We have compared our findings with those from other Latin American cohorts throughout the Discussion section (see lines 314–320 and 346–350).

- Discuss how private referral center context may influence results and generalizability, for example, differences in patient demographics, resource availability, or treatment protocols compared to public institutions

Thank you for your suggestion. The external validity of our results applies only to the private sector. Significant differences exist in the public sector. We have expanded the discussion of this topic (see lines 431-437).

6. PLOS authors have the option to publish the peer review history of their article (what does this mean?). If published, this will include your full peer review and any attached files.

Do you want your identity to be public for this peer review? For information about this choice, including consent withdrawal, please see our Privacy Policy.

Reviewer #1: No

Reviewer #2: Yes: Oluchi I Ndulue

Reviewer #3: Yes: Elizabeth B. Adedire

Questions embedded in the manuscript

How did you define their first episode of bacteremia? Since this is a referral center, are there patients who were already bacteremic on arrival? If yes, were they included in this study?

Only the first episode of bacteremia during the study period was considered. An episode of bacteremia was defined as a 28-day period beginning with the time of blood culture sampling. No patients were bacteremic on arrival; all were diagnosed during hospitalization, and all samples were processed in the center’s laboratory.

Do you mean at least one set of blood culture or just one of a set ?

It means at least one bottle of a set.

Were the sociodemographic and clinical characteristics of the transferred patients compared with that of the included patients to avoid bias. Eg could the transferred patient have different outcomes due to their characteristics?

The characteristics of patients lost to follow-up (n = 9) were not analyzed because they represented less than 3% of the total sample. We consider it unlikely that their exclusion substantially affected the overall characteristics of the study population.

Were both set taken from the same venous site or different sites?

Both sets were taken from different sites.

What of patients who were already on antibiotics for a different reason eg long term prophylaxis or UTI, were their samples incubated for longer than 24 hours?

Initially, all samples were cultured for at least 96 hours in media containing an antibiotic neutralizer. A second incubation was performed only for samples that showed initial growth.

Do you mean recurrent episodes of bacteremia during the same hospitalization or patients who already had history of recurrent bacteremia. One may have history of bacteremia in the past and a new first episode in a particular hospitalization. It will be helpful if you define what you mean by recurrent bacteremia and why you excluded them.

We excluded individuals with subsequent episodes of bacteremia; that is, if a patient developed a second or third episode months or years after the first, those episodes were excluded from the study. Recurrent episodes were excluded for statistical reasons. Including these recurrent events could introduce confounding bias and compromise the study's internal validity.

Did you do sensitivity analysis?

This question has already been answered.

What of other factors that could confound this outcome, eg neutropenia. Were they assessed/compared?

Yes, they were assessed. Besides the great proportion of elderly individuals and patients with a high burden of cancer, our cohort’s characteristics were not distant from those of other studies.

We hope the revisions satisfactorily address all concerns, and we would be grateful for your reconsideration of our manuscript for publication in PLOS ONE.

Sincerely,

Yosué I. Vera

---

## [Decision Letter · Decision Letter 1]

30 Apr 2026

28-Day All-Cause Mortality and Related Factors in Cancer Patients with Bacteremia at a Peruvian Referral Center

PONE-D-25-64720R1

Dear Dr. Vera,

We’re pleased to inform you that your manuscript has been judged scientifically suitable for publication and will be formally accepted for publication once it meets all outstanding technical requirements.

Kind regards,

Mabel Kamweli Aworh, DVM, MPH, PhD. FCVSN

Academic Editor

PLOS One

Additional Editor Comments (optional):

Reviewers' comments:

Reviewer's Responses to Questions

**Comments to the Author**

1. If the authors have adequately addressed your comments raised in a previous round of review and you feel that this manuscript is now acceptable for publication, you may indicate that here to bypass the “Comments to the Author” section, enter your conflict of interest statement in the “Confidential to Editor” section, and submit your "Accept" recommendation.

Reviewer #1: All comments have been addressed

Reviewer #2: All comments have been addressed

Reviewer #3: All comments have been addressed

2. Is the manuscript technically sound, and do the data support the conclusions?

Reviewer #1: Yes

Reviewer #2: Yes

Reviewer #3: Yes

3. Has the statistical analysis been performed appropriately and rigorously? 

Reviewer #1: Yes

Reviewer #2: Yes

Reviewer #3: Yes

4. Have the authors made all data underlying the findings in their manuscript fully available?

Reviewer #1: Yes

Reviewer #2: Yes

Reviewer #3: Yes

5. Is the manuscript presented in an intelligible fashion and written in standard English?

Reviewer #1: Yes

Reviewer #2: Yes

Reviewer #3: Yes

6. Review Comments to the Author

Reviewer #1: (No Response)

Reviewer #2: The authors did a great job reviewing and addressing the questions and suggestions from the reviewers.

Reviewer #3: (No Response)

7. PLOS authors have the option to publish the peer review history of their article (what does this mean?). If published, this will include your full peer review and any attached files.

Reviewer #1: No

Reviewer #2: **Yes:**Oluchi Ndulue

Reviewer #3: **Yes:**Elizabeth B. Adedire

---

## [Editor Report · Acceptance letter]

PONE-D-25-64720R1

PLOS One

Dear Dr. Vera,

I'm pleased to inform you that your manuscript has been deemed suitable for publication in PLOS One. Congratulations! Your manuscript is now being handed over to our production team.

Kind regards,

on behalf of

Dr. Mabel Kamweli Aworh

Academic Editor

PLOS One